# On the Tortuosity-Connectivity of Cement-Based Porous Materials

**Michiel Fenaux, Encarnacion Reyes** **, Jaime C. Gálvez \*****, Amparo Moragues**  **and Jesús Bernal** 

Escuela de Ingenieros de Caminos, Canales y Puertos, Universidad Politécnica de Madrid, 28040 Madrid, Spain;
michielfenaux@gmail.com (M.F.); encarnacion.reyes@upm.es (E.R.); amparo.moragues@upm.es (A.M.);
jmbc6784@hotmail.com (J.B.)
\* Correspondence: jaime.galvez@upm.es

**Abstract:** In this work, the transport equations of ionic species in concrete are studied. First, the equations at the porescale are considered, which are then averaged over a representative elementary volume. The so obtained transport equations at the macroscopic scale are thoroughly examined and each term is interpreted. Furthermore, it is shown that the tortuosity-connectivity does not slow the average speed of the ionic species down. The transport equations in the representative elementary volume are then compared with the equations obtained in an equivalent pore. Lastly, comparing Darcy's law and the Hagen–Poiseuille equation in a cylindrical equivalent pore, the tortuosity-connectivity parameter is obtained for four different concretes. The proposed model provides very good results when compared with the experimentally obtained chloride profiles for two additional concretes.

**Keywords:** tortuosity; microscale-macroscale; averaging; equivalent pore; ion transport; concrete

## 1. Introduction

The durability of concrete, which can be defined as its resistance to weathering action, chemical attacks and other degradation processes, is one of the most significant areas of research interest. Specifically, one of the most serious causes of durability problems affecting reinforcing steel is the introduction of chloride ions. The penetration of this aggressive ionic species into reinforced concrete induces steel corrosion and leads to premature deterioration of structures exposed to marine or de-icing salt environments [1,2]. As a result, most modern codes of practice have adopted strict maximum levels for chlorides permitted in concrete structures, among others the Spanish Standard of Structural Concrete EHE-08 [3]. Therefore, studying the processes involved in transport of ionic species in concrete is of utmost benefit.

Accurate prediction of the behaviour of concrete in different environments has been a challenge until now. The means by which ions penetrate into concrete at a given exposure time are complex and depend on several variables of the material, particularly on the moisture of the material. Bertolini [4] pointed out that, among the different penetration mechanisms, four should be considered, namely diffusion, migration, absorption/capillary suction and permeation. An enormous variety of mathematical models has been proposed over the years with the main goal to explain and predict the penetration of ionic species into concrete [5–10]. The most simple models are generally based on the diffusion equation with an apparent diffusion coefficient which depends on the material as well as on the concentration of the ionic species [11–14]. The description of the various processes involved, requires the consideration of various transport mechanisms coupled with the effect of the interaction between the ion and the pore surface, which is very challenging. Despite this, some authors have recently proposed advanced approaches aiming at modelling the features of chloride transport in difficult and variable exposure conditions. Thus, Song et al. [15] studied how different types of cation influence the diffusive process of

chloride ions in concrete. Ožbolt et al. [16] proposed a 3D chemo-hygro-thermo mechanical model for reinforced concrete to simulate physical processes related to corrosion of steel bars. In a more recent work, Zang et al. [17] have proposed a model that couples the ionic diffusion process with the concrete microstructure evolution.

The problem is further complicated when dealing with heterogeneous and porous materials, such as concrete. Indeed, the complex geometry of the porous network of cement-based materials is tortuous in nature. The geometry of the pathways of this materials for the penetration of aggressive agents is very complicated, even further due to its heterogeneity and particularly the higher porosity of the interface zone can facilitate the ingress of external aggressive agents and the development of deleterious chemical reactions. Bourdette et al. [18] found a notably higher effective diffusion coefficient of chloride ions in the interface zone (6 to12 times greater) than in the bulk cement paste. So, the detailed modelling of cement-based materials microstructure requires to consider different levels of approximation, which would greatly complicate the modelling. In addition, micro-models consider a large number of parameters which are usually difficult to calibrate through specific tests [17]. Seeking to more practical approaches, in previous works, a constant tortuosity factor has been adopted which assumes that all the effects arising from pore orientation, connectivity, size variation, etc., can be encompassed by a mean value valid for all pore sizes [19–21]. This is obviously a gross assumption since it depends on the implicit assumption that the effects of pore geometry and structure are the same for all pore sizes, even so, if the pore structure is characterized in detail, a sufficiently accurate aproach can be made. Ahmad [19] studied the relationship between permeability and tortuosity, based on the fact that both depend upon the pore structure and its interconnectivity.

This work focuses on the influence of the geometric complexity of the porous network on the different mass fluxes. First, the equations at the porescale are considered, which are then averaged over a representative elementary volume. This procedure shows the origin of the tortuosity-connectivity parameter used for modelling ionic transport in porous media. An explicit expression for the tortuosity-connectivity parameter is finally obtained by comparing Darcy's law and the Hagen–Poiseuille equation in a cylindrical equivalent pore. This expression can be written as a function of the pore radii, or more conveniently, as a function of the pore water content.

## 2. General Considerations

In order to determine the differential equations which govern the transport problem at the macroscopic scale, the equations at the pore scale are integrated in a representative elementary volume (REV). The REV $\Omega$ is assumed to contain three phases, namely the liquid phase (the pore water), the gaseous phase (air) and the solid phase (concrete). Furthermore, it is assumed that the ionic species are transported solely through the liquid phase, and the solid phase is assumed to be inert. The REV can be seen as a whole of multiple elementary volumes. In this work, only homogeneous materials are considered, so that each elementary volume has the same properties. The characteristic length of the elementary volume is denoted as $l$, while the characteristic length of the REV is denoted as $L$. Furthermore, the size of the REV is chosen such that $\epsilon = l/L << 1$. A representation of one elementary volume, projected onto a 2D plane, is depicted in Figure 1. The volume of the liquid phase is denoted as $\Omega_l$, while the boundaries between the phases $i$ and $j$ are denoted as $A_{ij}$. Finally, the unit vectors $\boldsymbol{n}_{ij}$ are normal to the $i - j$ interface pointing from the $i$-phase to the $j$-phase.

Throughout this work, the following notations are adopted:

$$\langle \ \cdot \ \rangle_\Omega = \frac{1}{\Omega} \int_\Omega \cdot \, dV \tag{1}$$

$$\langle \ \cdot \ \rangle_{\Omega_l} = \frac{1}{\Omega_l} \int_{\Omega_l} \cdot \, dV \tag{2}$$

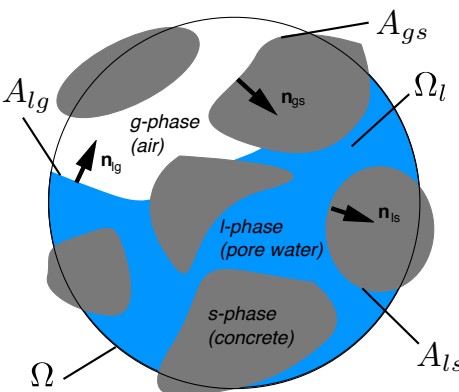

**Figure 1.** Elementary volume.

Note that Equations (1) and (2) are related as follows:

$$\langle \, \cdot \, \rangle_\Omega = \phi_l \langle \, \cdot \, \rangle_{\Omega_l} \tag{3}$$

where $\phi_l = \Omega_l / \Omega$ is the pore water content.

Furthermore, let $u$ be a quantity related to the $l$-phase and consider the following decomposition [22]:

$$u = \langle u \rangle_{\Omega_l} + \tilde{u} \tag{4}$$

$$\langle \tilde{u} \rangle_{\Omega_l} = 0 \tag{5}$$

where $\tilde{u}$ is the deviation of $u$ with respect to the average $\langle u \rangle_{\Omega_l}$. From Equation (5), the following relations may be deduced:

$$\langle \langle u \rangle_{\Omega_l} \rangle_{\Omega_l} = \langle u \rangle_{\Omega_l} \tag{6}$$

$$\tilde{\tilde{u}} = \tilde{u} \tag{7}$$

Equation (6) states that the average of the average value of $u$ in $\Omega_l$ is simply the average itself, while Equation (7) states that the deviation of $\tilde{u}$ is equal to $\tilde{u}$.

Moreover, the following theorems are used throughout this work [23–25] :

$$\langle \frac{\partial u}{\partial t} \rangle_\Omega = \frac{\partial}{\partial t} \langle u \rangle_\Omega - \frac{1}{\Omega} \int_{A_{li}} u \boldsymbol{v}_{li} \cdot \boldsymbol{n}_{li} dA \qquad \textit{Reynolds} \tag{8}$$

$$\langle \nabla u \rangle_\Omega = \nabla \langle u \rangle_\Omega + \frac{1}{\Omega} \int_{A_{li}} u \boldsymbol{n}_{li} dA \qquad \textit{Averaging} \tag{9}$$

$$\langle \nabla . \boldsymbol{J} \rangle_\Omega = \nabla . \langle \boldsymbol{J} \rangle_\Omega + \frac{1}{\Omega} \int_{A_{li}} \boldsymbol{J} \cdot \boldsymbol{n}_{li} dA \qquad \textit{Averaging} \tag{10}$$

where $\boldsymbol{J}$ is a vector quantity related to the $l$-phase, $i$ refers to the $g$-phase and the $s$-phase and $\boldsymbol{v}_{li}$ is the velocity vector of the $l - i$ interface.

Equations (8)–(10) referred to the volume $\Omega_l$ read:

$$\langle \frac{\partial u}{\partial t} \rangle_{\Omega_l} = \frac{1}{\phi_l} \frac{\partial}{\partial t} \left( \phi_l \langle u \rangle_{\Omega_l} \right) - \frac{1}{\Omega_l} \int_{A_{li}} u \boldsymbol{v}_{li} \cdot \boldsymbol{n}_{li} dA \tag{11}$$

$$\langle \nabla u \rangle_{\Omega_l} = \frac{1}{\phi_l} \nabla \left( \phi_l \langle u \rangle_{\Omega_l} \right) + \frac{1}{\Omega_l} \int_{A_{li}} u \boldsymbol{n}_{li} dA \tag{12}$$

$$\langle \nabla . \boldsymbol{J} \rangle_{\Omega_l} = \frac{1}{\phi_l} \nabla . \left( \phi_l \langle \boldsymbol{J} \rangle_{\Omega_l} \right) + \frac{1}{\Omega_l} \int_{A_{li}} \boldsymbol{J} \cdot \boldsymbol{n}_{li} dA \tag{13}$$

Applying Equations (8) and (9) to $u = 1$, the following lemmas are obtained:

$$\frac{1}{\phi_l}\frac{\partial \phi_l}{\partial t} = \frac{1}{\Omega_l}\int_{A_{li}} \boldsymbol{v}_{li}\cdot \boldsymbol{n}_{li}dA \tag{14}$$

$$\frac{1}{\phi_l}\nabla \phi_l = -\frac{1}{\Omega_l}\int_{A_{li}} \boldsymbol{n}_{li}dA \tag{15}$$

Using Equations (14) and (15), Equations (11) and (12) can be rewritten as:

$$\langle \frac{\partial u}{\partial t}\rangle_{\Omega_l} = \frac{\partial}{\partial t}\langle u\rangle_{\Omega_l} - \frac{1}{\Omega_l}\int_{A_{li}} \tilde{u}\boldsymbol{v}_{li}\cdot \boldsymbol{n}_{li}dA \tag{16}$$

$$\langle \nabla u\rangle_{\Omega_l} = \nabla \langle u\rangle_{\Omega_l} + \frac{1}{\Omega_l}\int_{A_{li}} \tilde{u}\boldsymbol{n}_{li}dA \tag{17}$$

## 3. The Transport Equations at the Pore Scale

Denoting the concentration, mass flux and reaction terms of species $i$ by $u_i$ (kg/m$^3$), $\boldsymbol{J^i}$ (kg/m$^2$/s) and $r_i$ (kg/m$^3$/s), respectively, the transport equation at the pore scale is formulated as follows:

$$\frac{\partial u_i}{\partial t} + \nabla.\boldsymbol{J^i} = r_i \tag{18}$$

where each term is expressed in units of mass per volume of pore water per second (kg/m$^3$/s).

The total mass flux is the sum of the diffusive mass flux $\boldsymbol{J}_D^i$, the advective mass flux $\boldsymbol{J}_A^i$, the mass flux related to migration $\boldsymbol{J}_M^i$ and the mass fluxes due to temperature gradients and chemical activity $\boldsymbol{J}_T^i$ [5]:

$$\boldsymbol{J^i} = \boldsymbol{J}_D^i + \boldsymbol{J}_A^i + \boldsymbol{J}_M^i + \boldsymbol{J}_T^i \tag{19}$$

$$\boldsymbol{J}_D^i = -D_i\nabla u_i \tag{20}$$

$$\boldsymbol{J}_A^i = \boldsymbol{a}u_i \tag{21}$$

$$\boldsymbol{J}_M^i = -\frac{z_iF}{RT}D_iu_i\nabla\Phi \tag{22}$$

$$\boldsymbol{J}_T^i = -D_iu_i\gamma_i^{-1}\nabla\gamma_i - D_iu_i\ln(a_i)T^{-1}\nabla T \tag{23}$$

where $D$ (m$^2$/s) is the diffusion coefficient at infinite dilution and solely depends on temperature, $\boldsymbol{a}$ (m/s) the advection velocity (i.e., the velocity of the pore water), $z$ the valence, $F$ (C/mol) the Faraday constant, $R$ (J/mol/K) the gas constant, $T$ (K) the temperature, $\Phi$ (V) the electric potential, $\gamma$ (m$^3$/kg) the activity coefficient and $a$ the chemical activity. An explicit expression for the advection velocity $\boldsymbol{a}$ will be given later.

Substitution of Equations (19)–(23) into Equation (18) yields:

$$\frac{\partial u_i}{\partial t} + \nabla.(\boldsymbol{a}_iu_i) = \nabla.\Big(D_i\nabla u_i + \frac{z_iF}{RT}D_iu_i\nabla\Phi + $$
$$D_iu_i\gamma_i^{-1}\nabla\gamma_i + D_iu_i\ln(a_i)T^{-1}\nabla T\Big) + r_i \tag{24}$$

## 4. The Transport Equations at the Macroscopic Scale

In order to obtain the equations at the macroscopic scale, the microscopic transport equation needs to be integrated in the REV. Thus, Equation (24) is first integrated in the elementary volume $\Omega$:

$$\langle \frac{\partial u_i}{\partial t}\rangle_{\Omega} + \langle \nabla.\boldsymbol{J^i}\rangle_{\Omega} = \langle r_i\rangle_{\Omega} \tag{25}$$

In the following, each term will be treated separately.

### 4.1. The Time Derivative

The average of the time derivative is obtained by applying Reynolds' transport theorem (Equation (8)) to the first term on the left-hand side of Equation (24):

$$\langle \frac{\partial u_i}{\partial t} \rangle_\Omega = \frac{\partial}{\partial t} \langle u_i \rangle_\Omega - \frac{1}{\Omega} \int_{A_{lg}} u_i v_{lg} \cdot n_{lg} dA$$
$$- \frac{1}{\Omega} \int_{A_{ls}} u_i v_{ls} \cdot n_{ls} dA \tag{26}$$

It should be noted that the velocity of the $l - s$ interface is not necessarily equal to zero. Indeed, if the ionic species react to the cement matrix, the boundary between the $l$-phase and the $s$-phase may be deformed. However, for the sake of simplicity, it is assumed that this deformation does not alter significantly the transport of the ionic species, which simplifies Equation (26) to:

$$\langle \frac{\partial u_i}{\partial t} \rangle_\Omega = \frac{\partial}{\partial t} \langle u_i \rangle_\Omega - \frac{1}{\Omega} \int_{A_{lg}} u_i v_{lg} \cdot n_{lg} dA \tag{27}$$

### 4.2. The Divergence of the Mass Fluxes

The average of the divergence of the total mass flux is obtained by applying the averaging theorem (Equation (10)) to the second term of the left-hand side of Equation (18):

$$\langle \nabla . J^i \rangle_\Omega = \nabla . \langle J^i \rangle_\Omega + \frac{1}{\Omega} \int_{A_{lg}} J^i \cdot n_{lg} dA$$
$$+ \frac{1}{\Omega} \int_{A_{ls}} J^i \cdot n_{ls} dA \tag{28}$$

Since the ionic species are assumed to be transported solely through the $l$-phase, and assuming that the temperature is constant within the elementary volume, the following boundary conditions can be applied:

$$J_D^i \cdot n_{lg} = 0 = J_D^i \cdot n_{ls} \tag{29}$$
$$J_A^i \cdot n_{ls} = 0 \tag{30}$$
$$J_M^i \cdot n_{lg} = 0 = J_M^i \cdot n_{ls} \tag{31}$$
$$J_T^i \cdot n_{lg} = 0 = J_T^i \cdot n_{ls} \tag{32}$$

The assumption of a constant temperature within the elementary volume, which will be adopted in the rest of this paper, can be justified by the high thermal conductivity of concrete when compared with the slow penetration of the ionic species and the pore water [5]. Moreover, it was found in [5] that temperature plays an important role in the transport of ionic species in concrete, while temperature gradients do not.

Now, the average of the divergence of the mass fluxes are obtained by means of Equation (28) and the boundary conditions (29)–(32):

$$\langle \nabla . J_D^i \rangle_\Omega = -D_i \langle \nabla . (\nabla u_i) \rangle_\Omega = -D_i \nabla . \langle \nabla u_i \rangle_\Omega \tag{33}$$

$$\langle \nabla . J_A^i \rangle_\Omega = \langle \nabla . (a u_i) \rangle_\Omega = \nabla . \langle a u_i \rangle_\Omega + \frac{1}{\Omega} \int_{A_{lg}} u_i a \cdot n_{lg} dA \tag{34}$$

$$\langle \nabla . J_M^i \rangle_\Omega = -\frac{z_i F}{RT} D_i \langle \nabla . (u_i \nabla \Phi) \rangle_\Omega = -\frac{z_i F}{RT} D_i \nabla . \langle u_i \nabla \Phi \rangle_\Omega \tag{35}$$

$$\langle \nabla . J_T^i \rangle_\Omega = -D_i \langle \nabla . \left( u_i \gamma_i^{-1} \nabla \gamma_i + u_i \ln(a_i) T^{-1} \nabla T \right) \rangle_\Omega$$
$$= -D_i \nabla . \langle u_i \gamma_i^{-1} \nabla \gamma_i + u_i \ln(a_i) T^{-1} \nabla T \rangle_\Omega \tag{36}$$

*4.3. The Transport Equations at the Macroscopic Scale*

Substituting Equation (27) and Equations (33)–(36) into Equation (25), the transport equations at the macroscopic scale are obtained:

$$\frac{\partial}{\partial t}\langle u_i\rangle_\Omega + \nabla.\langle au_i\rangle_\Omega + \frac{1}{\Omega}\int_{A_{lg}} u_i\left(a - v_{lg}\right)\cdot n_{lg}dA$$

$$= D_i\nabla.\langle\nabla u_i\rangle_\Omega + \frac{z_iF}{RT}D_i\nabla.\langle u_i\nabla\Phi\rangle_\Omega$$

$$+ D_i\nabla.\langle u_i\gamma_i^{-1}\nabla\gamma_i + u_i\ln(a_i)T^{-1}\nabla T\rangle_\Omega + \langle r_i\rangle_\Omega \qquad (37)$$

The velocity $v_{lg}$ represents the total velocity of the $l-g$ interface, while $a$ at the boundary is the velocity of the interface due to the movement of the $l$-phase. In other words, unlike $a$, $v_{lg}$ accounts for the velocity of the interface due to the evaporation and condensation of the pore water. The velocity $v_{lg}$ can thus be interpreted as the sum of the velocity at the $l-g$ interface $a|_{lg}$ and the velocity of the boundary due to evaporation and condensation $v_{lg,vap}$:

$$v_{lg} = a|_{lg} + v_{lg,vap} \qquad (38)$$

Combining Equations (14) and (38), the variations of the pore water content with time can be expressed as follows:

$$\frac{\partial\phi_l}{\partial t} = \frac{1}{\Omega}\int_{A_{lg}} a\cdot n_{lg}dA + \frac{1}{\Omega}\int_{A_{lg}} v_{lg,vap}\cdot n_{lg}dA \qquad (39)$$

where the second surface integral of Equation (39) represents the change of $\phi_l$ with time due to evaporation and condensation of the pore water. Mainguy et al. [26] showed that when concrete is subjected to drying processes the pore water is evaporated at the surface of the material, and the evaporation within the material can be ignored. Therefore, the effects of evaporation and condensation within the elementary volume are neglected, so that $v_{lg,vap} = 0$.

Equation (37) can now be rewritten as:

$$\frac{\partial}{\partial t}\langle u_i\rangle_\Omega + \nabla.\langle au_i\rangle_\Omega = D_i\nabla.\langle\nabla u_i\rangle_\Omega + \frac{z_iF}{RT}D_i\nabla.\langle u_i\nabla\Phi\rangle_\Omega +$$

$$D_i\nabla.\langle u_i\gamma_i^{-1}\nabla\gamma_i + u_i\ln(a_i)T^{-1}\nabla T\rangle_\Omega + \langle r_i\rangle_\Omega \qquad (40)$$

or with respect to volume $\Omega_l$:

$$\frac{\partial}{\partial t}\left(\phi_l\langle u_i\rangle_{\Omega_l}\right) + \nabla.(\phi_l\langle au_i\rangle_\Omega)$$

$$= D_i\nabla.\left(\phi_l\langle\nabla u_i\rangle_{\Omega_l}\right) + \frac{z_iF}{RT}D_i\nabla.\left(\phi_l\langle u_i\nabla\Phi\rangle_{\Omega_l}\right)$$

$$+ D_i\nabla.\left(\phi_l\langle u_i\gamma_i^{-1}\nabla\gamma_i + u_i\ln(a_i)T^{-1}\nabla T\rangle_{\Omega_l}\right) + \phi_l\langle r_i\rangle_{\Omega_l} \qquad (41)$$

*4.4. The Mass Fluxes*

Explicit expressions for the mass fluxes are given in Equations (20)–(23), except for the advective mass flux, which depends on the advection velocity $a$. The advection velocity is not known a priori and depends strongly on the pore water content $\phi_l$. In this work, $a$ is modelled by means of Darcy's law, based on research work [5]:

$$a = \frac{k_l}{\nu_l}\nabla p_c \qquad (42)$$

where $k_l$ (m$^2$) is the permeability of the porous medium, $\nu_l$ (Pa·s) is the dynamic viscosity of the pore water, and $p_c$ (Pa) is the capillary pressure. The permeability of the porous medium relative to the pore water depends on the pore water content, the dynamic viscosity of the

pore water depends on the concentration of the ionic species and on temperature, and the capillary pressure depends on the pore water content, temperature and porosity [5]. As the porosity can change due to chemical reactions of the ionic species with the cement matrix, and those reactions depend directly on the concentration of the ions, the capillary pressure may be expressed as a function of the concentration, rather than the porosity.

The advective mass flux (21) at the pore scale now reads:

$$J_A^i = u_i \frac{k_l}{\nu_l} \nabla p_c \tag{43}$$

The averages of the mass fluxes are obtained by applying the average theorem (Equation (9)) and the decomposition, as shown in Equation (4) to the fluxes defined in Equations (20), (22)–(23) and (43):

$$\langle J_D^i \rangle_\Omega = - D_i \left( \nabla \langle u_i \rangle_\Omega + \frac{1}{\Omega} \int_{A_{lj}} u_i \boldsymbol{n}_{lj} dA \right) \tag{44}$$

$$\langle J_A^i \rangle_\Omega = \langle u_i \rangle_\Omega \phi_l^{-1} \frac{k_l}{\nu_l} \left( \nabla \langle p_c \rangle_\Omega + \frac{1}{\Omega} \int_{A_{lj}} p_c \boldsymbol{n}_{lj} dA \right)$$
$$+ \frac{k_l}{\nu_l} \langle \tilde{u}_i \nabla p_c \rangle_\Omega \tag{45}$$

$$\langle J_M^i \rangle_\Omega = - \frac{z_i F}{RT} D_i \langle u \rangle_{\Omega_l} \left( \nabla \langle \Phi \rangle_\Omega + \frac{1}{\Omega} \int_{A_{lj}} \Phi \boldsymbol{n}_{lj} dA \right)$$
$$- \frac{z_i F}{RT} D_i \langle \tilde{u} \nabla \Phi \rangle_\Omega \tag{46}$$

$$\langle J_T^i \rangle_\Omega = - D_i \langle u_i \rangle_{\Omega_l} \left( \nabla \langle \ln \gamma_i \rangle_\Omega + \langle \ln a_i \rangle_{\Omega_l} \nabla \langle \ln T \rangle_\Omega \right.$$
$$+ \frac{1}{\Omega} \int_{A_{lj}} \left( \ln \gamma_i + \langle \ln a_i \rangle_{\Omega_l} \ln T \right) \boldsymbol{n}_{lj} dA \right)$$
$$- D_i \langle \tilde{u}_i \nabla \ln \gamma_i + \left( \tilde{u}_i \ln a_i + \langle u_i \rangle_{\Omega_l} \widetilde{\ln a_i} \right) \nabla \ln T \rangle_\Omega \tag{47}$$

where the subscript $j$ refers to the $g$- and $s$-phases.

Note that, in Equation (44), there is no need to calculate the average of the diffusion coefficient. Indeed, $D$ solely depends on temperature which is assumed to be constant in the elementary volume. Furthermore, it should be noted that Equation (45) was obtained by assuming that the deviations of $u_i$ within the elementary volume are sufficiently small so that the dynamic viscosity $\nu_l$ can be assumed constant in the elementary volume. This approximation can be expressed as follows:

$$\nu_l(u_i, T) = \nu_l(\langle u_i \rangle_{\Omega_l} + \tilde{u}, T) \simeq \nu_l(\langle u_i \rangle_{\Omega_l}, T) \tag{48}$$

Substitution of the expressions (44)–(47) into Equation (40) yields the final transport equation in the elementary volume:

$$\frac{\partial}{\partial t} \langle u_i \rangle_\Omega + \nabla. \left( \langle u_i \rangle_\Omega \phi_l^{-1} \frac{k_l}{\nu_l} \left( \nabla \langle p_c \rangle_\Omega + \frac{1}{\Omega} \int_{A_{lj}} p_c \boldsymbol{n}_{lj} dA \right) \right)$$
$$= D_i \nabla. \left( \nabla \langle u_i \rangle_\Omega + \frac{1}{\Omega} \int_{A_{li}} u_i \boldsymbol{n}_{lj} dA \right)$$
$$+ \frac{z_i F}{R} \nabla. \left( D_i T^{-1} \langle u_i \rangle_{\Omega_l} \left[ \nabla \langle \Phi \rangle_\Omega + \frac{1}{\Omega} \int_{A_{lj}} \Phi \boldsymbol{n}_{lj} dA \right] \right)$$

$$+ \nabla . \left( D_i \langle u_i \rangle_{\Omega_l} \left[ \nabla \langle \ln \gamma_i \rangle_\Omega + \langle \ln a_i \rangle_{\Omega_l} \nabla \langle \ln T \rangle_\Omega \right. \right.$$

$$\left. \left. + \frac{1}{\Omega} \int_{A_{lj}} \left( \ln \gamma_i + \langle \ln a_i \rangle_{\Omega_l} \ln T \right) \boldsymbol{n}_{lj} dA \right] \right)$$

$$+ \frac{z_i F}{R} \nabla . \left( D_i T^{-1} \langle \tilde{u}_i \nabla \Phi \rangle_\Omega \right) + \nabla . (D_i \langle \tilde{u}_i \nabla \ln \gamma_i \rangle_\Omega)$$

$$+ \nabla . \left( D_i \langle \left[ \tilde{u}_i \ln a_i + \langle u_i \rangle_{\Omega_l} \widetilde{\ln a_i} \right] \nabla \ln T \rangle_\Omega \right)$$

$$- \nabla . \left( \frac{k_l}{\nu_l} \langle \tilde{u}_i \nabla p_c \rangle_\Omega \right) + \langle r_i \rangle_\Omega \tag{49}$$

*4.5. The Dispersion Terms*

In this subsection, it is shown that the dispersion terms of Equation (49) can be ignored in the REV. To that end, consider a quantity *m* related to the *l*-phase. According to Equation (4), *m* can be decomposed as:

$$m = \langle m \rangle_{\Omega_l} + \tilde{m} \tag{50}$$

Since the microscopic quantity *m* may vary significantly over a distance equal to the characteristic length *l*, the magnitude of the deviation of *m* with respect to the average $\langle m \rangle_{\Omega_l}$ can be estimated as follows:

$$\| \tilde{m} \| \simeq l \, \| \nabla m \| \tag{51}$$

Furthermore, the magnitude of a quantity at the macroscale only varies significantly over distances larger than *L*, which can be expressed as:

$$\| \nabla . \langle \cdot \rangle_\Omega \| \simeq L^{-1} \, \| d \langle \cdot \rangle_\Omega \| \tag{52}$$

Applying those approximations to Equation (49), and noting that the diffusion coefficient can change within the REV due to temperature variations, the final transport equation in the REV is obtained:

$$\frac{\partial}{\partial t} \langle u_i \rangle_{\hat{\Omega}} + \nabla . \left( \langle u_i \rangle_{\hat{\Omega}} \phi_l^{-1} \frac{k_l}{\nu_l} \left[ \nabla \langle p_c \rangle_{\hat{\Omega}} + \frac{1}{\hat{\Omega}} \int_{\hat{A}_{lj}} p_c \boldsymbol{n}_{lj} dA \right] \right)$$

$$= \nabla . \left( D_i \left[ \nabla \langle u_i \rangle_{\hat{\Omega}} + \frac{1}{\hat{\Omega}} \int_{\hat{A}_{lj}} u_i \boldsymbol{n}_{lj} dA \right] \right)$$

$$+ \frac{z_i F}{R} \nabla . \left( D_i T^{-1} \langle u_i \rangle_{\hat{\Omega}} \left[ \nabla \langle \Phi \rangle_{\hat{\Omega}} + \frac{1}{\hat{\Omega}} \int_{\hat{A}_{lj}} \Phi \boldsymbol{n}_{lj} dA \right] \right)$$

$$+ \nabla . \left( D_i \langle u_i \rangle_{\hat{\Omega}} \left[ \nabla \langle \ln \gamma_i \rangle_{\hat{\Omega}} + \langle \ln a_i \rangle_{\hat{\Omega}_l} \nabla \langle \ln T \rangle_{\hat{\Omega}} \right. \right.$$

$$\left. \left. + \frac{1}{\hat{\Omega}} \int_{\hat{A}_{lj}} \left( \ln \gamma_i + \langle \ln a_i \rangle_{\hat{\Omega}_l} \ln T \right) \boldsymbol{n}_{lj} dA \right] \right) + \langle r_i \rangle_{\hat{\Omega}} \tag{53}$$

where $\hat{\Omega}$, $\hat{\Omega}_l$ and $\hat{A}$ are related to the REV.

*4.6. The Surface Integrals*

From Equation (53), it may be observed that all the mass fluxes in the REV have the form:

$$\langle J^i \rangle_{\hat{\Omega}} = f \left( \nabla \langle \, \cdot \, \rangle_{\hat{\Omega}} + \frac{1}{\hat{\Omega}} \int_{\hat{A}_{lj}} \, \cdot \, \boldsymbol{n}_{lj} dA \right) \tag{54}$$

where $J^i$ is a mass flux of species *i* and *f* is a scalar.

In order to interpret the terms on the right-hand side of Equation (54), consider an elementary volume with only one straight cylindrical pore. The REV can then be thought of as a volume with a single cylindrical pore. In such case, the second term on the right-hand side of Equation (54) is zero. Therefore, the first term can be interpreted as the mass flux through the cylindrical pore. The second term corrects the mass flux by accounting for the complex geometry of the porous medium as shown below for the diffusion flux. For the sake of simplicity, the fluxes related to migration and chemical activity are neglected [5,27].

### 4.6.1. The Surface Integrals at the Pore Scale

The second term on the right-hand side of Equation (54) is often associated with the tortuosity-connectivity of the porous medium. Generally, the more tortuous the porous material is, the slower is the penetration of the ionic species. However, it should be noted that the term which accounts for the geometry of the porous network does not slow down the speed of the ionic species. To illustrate this, the simple diffusion equation at the pore scale for some ionic species $i$ is considered:

$$\frac{\partial u_i}{\partial t} = D_i \Delta u_i \tag{55}$$

where $\Delta$ is the Laplace operator.

Two kinds of pores are analyzed, namely a straight cylindrical pore and a tortuous pore. The dimensions of the pores were chosen such that the length and volume of each pore were equal.

A concentration of $u_i = 1 \text{ kg/m}^3$ was imposed on the left boundary of the pores. The remaining boundaries are taken to be the $l - s$ interface, where a zero mass flux is imposed. With the aim of nondimensionalizing Equation (56) the value of the diffusion coefficient was chosen such that:

$$\frac{t_{sim} D_i}{d^2} = 1 \tag{56}$$

where $t_{sim}$ refers to the total simulation time, and $d$ the length of the pore.

The dimensionless form of Equation (55) was solved by means of the finite element method. The results at time $t = 0.38 t_{sim}$ are shown in Figures 2 and 3. It may be observed that the ionic species reach the right boundary faster in the tortuous pore than in the cylindrical pore. This shows that tortuosity-connectivity increases the average speed of the ions.

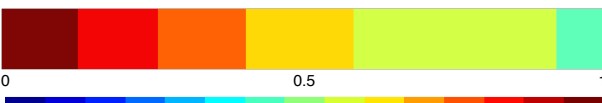

**Figure 2.** Concentration in the cylindrical pore.

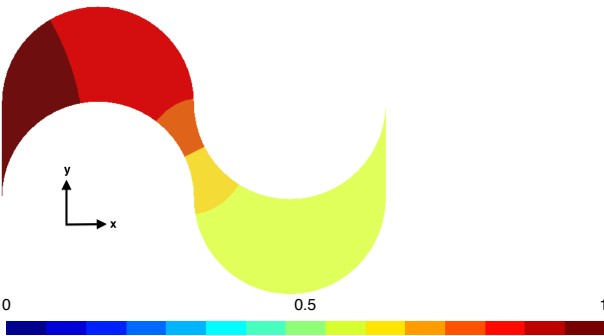

**Figure 3.** Concentration in a tortuous pore.

In conclusion, the tortuous pore may be represented by means of a straight pore of equal volume, orientated in the direction of the net mass flux, and of length equal to the length of the tortuous pore projected onto the direction of the net mass flux. The latter

will be referred to as the equivalent pore. The mass flux in the tortuous pore can then be expressed as the sum of the diffusive flux in the equivalent pore and a term in the form of a surface integral which accounts for the geometry of the tortuous pore, as stated in Equation (54).

### 4.6.2. The Surface Integrals in the REV

In the REV, the porous network can be significantly more complex than the previously defined tortuous pore. An equivalent pore of length $L$ and volume $\phi_l$ can be constructed, oriented in the direction of the net mass flux. The mass flux averaged over the equivalent pore is then equal to the mass flux averaged over that part of the REV which contains pore water:

$$\langle J_D^i \rangle_{\hat{\Omega}_{eq}} = -D_i \left( \nabla \langle u_i \rangle_{\hat{\Omega}_l} + \frac{1}{\hat{\Omega}_l} \int_{\hat{A}_{lj}} \tilde{u}_i \boldsymbol{n}_{lj} dA \right) \tag{57}$$

where $\hat{\Omega}_{eq}$ refers to the volume of the equivalent pore.

Note that the average of the mass flux in the equivalent pore, as defined in Equation (57), accounts for the tortuosity-connectivity of that part of the elementary volume which contains pore water. The last term reduces the mass flux in the direction parallel to the equivalent pore. Furthermore, the direction of the equivalent pore is parallel to the direction of $\nabla \langle u_i \rangle_{\Omega_l}$. Taking this into account, Equation (57) can be approximated as:

$$\langle J_D^i \rangle_{\hat{\Omega}_{eq}} \simeq -D_i \left( \nabla \langle u_i \rangle_{\hat{\Omega}_l} - \sigma \nabla \langle u_i \rangle_{\hat{\Omega}_l} \right)$$
$$= -D_i \left( [1-\sigma] \nabla \langle u_i \rangle_{\hat{\Omega}_l} \right) \tag{58}$$

where $\sigma$ is a dimensionless positive function which accounts for the reduction of the mass flux in the direction of the equivalent pore.

Finally, the average of the mass flux in the REV is obtained by multiplying Equation (58) by $\phi_l$:

$$\langle J_D^i \rangle_{\hat{\Omega}} = -\phi_l D_i [1-\sigma] \nabla \langle u_i \rangle_{\hat{\Omega}_l} \tag{59}$$

### 4.6.3. The Final Transport Equation in the Equivalent Pore

Based on the results obtained in Section 4.6.2, the transport equation in the equivalent pore related to the REV is given by:

$$\frac{\partial}{\partial t} \left( \phi_l \langle u_i \rangle_{\hat{\Omega}_l} \right) + \nabla . \left( \langle u_i \rangle_{\hat{\Omega}_l} \frac{k_l}{\nu_l} [1-\sigma] \nabla \langle p_c \rangle_{\hat{\Omega}_l} \right)$$
$$= \nabla . \left( \phi_l D_i [1-\sigma] \nabla \langle u_i \rangle_{\hat{\Omega}_l} \right)$$
$$+ \frac{z_i F}{R} \nabla . \left( \phi_l D_i T^{-1} \langle u_i \rangle_{\hat{\Omega}_l} [1-\sigma] \nabla \langle \Phi \rangle_{\hat{\Omega}_l} \right)$$
$$+ \nabla . \left( \phi_l D_i \langle u_i \rangle_{\hat{\Omega}_l} [1-\sigma] \nabla \langle \ln \gamma_i \rangle_{\hat{\Omega}_l} \right)$$
$$+ \nabla . \left( \phi_l D_i \langle u_i \rangle_{\hat{\Omega}_l} \langle \ln a_i \rangle_{\hat{\Omega}_l} [1-\sigma] \nabla \langle \ln T \rangle_{\hat{\Omega}_l} \right)$$
$$+ \phi_l \langle r_i \rangle_{\hat{\Omega}_l} \tag{60}$$

Finally, defining the tortuosity-connectivity parameter as $\tau = 1 - \sigma$, the final transport equation is obtained:

$$\frac{\partial}{\partial t}\left(\phi_l \langle u_i \rangle_{\hat{\Omega}_l}\right) + \nabla.\left(\langle u_i \rangle_{\hat{\Omega}_l}\frac{k_l}{\nu_l}\tau \nabla \langle p_c \rangle_{\hat{\Omega}_l}\right)$$

$$= \nabla.\left(\phi_l D_i \tau \nabla \langle u_i \rangle_{\hat{\Omega}_l}\right)$$

$$+ \frac{z_i F}{R}\nabla.\left(\phi_l D_i T^{-1}\langle u_i \rangle_{\hat{\Omega}_l}\tau \nabla \langle \Phi \rangle_{\hat{\Omega}_l}\right)$$

$$+ \nabla.\left(\phi_l D_i \langle u_i \rangle_{\hat{\Omega}_l}\tau \nabla \langle \ln \gamma_i \rangle_{\hat{\Omega}_l}\right)$$

$$+ \nabla.\left(\phi_l D_i \langle u_i \rangle_{\hat{\Omega}_l}\langle \ln a_i \rangle_{\hat{\Omega}_l}\tau \nabla \langle \ln T \rangle_{\hat{\Omega}_l}\right)$$

$$+ \phi_l \langle r_i \rangle_{\hat{\Omega}_l} \tag{61}$$

## 5. Determining the Tortuosity-Connectivity Parameter

### 5.1. Darcy versus Hagen–Poiseuille Flows

The tortuosity-connectivity parameter can be determined by comparing the Darcy flux with the Hagen–Poiseuille fluxes in a cylindrical pore. The following analysis, it should be noted, is valid only for homogeneous materials. Consider a homogeneous material, partially saturated with pore water which is assumed to be continuously and uniformly distributed. Then the equivalent pore related to the porous network which contains pore water can be chosen to be a cylindrical pore of constant section. In that case, the Darcy and Hagen–Poiseuille fluxes must coincide, which leads to the expression of the tortuosity-connectivity as a function of the permeability of the porous medium.

Based on the obtained results, the Darcy flux in the equivalent pore can be expressed as:

$$\langle J^i_{Darcy} \rangle_{\hat{\Omega}} = -\frac{\hat{k}_l}{\nu_l}\nabla \langle p_l \rangle_{\hat{\Omega}} \tag{62}$$

where $\hat{k}_l = \tau k_l$ is now a macroscopic function which describes the permeability of the complex and tortuous porous network.

The Hagen–Poiseuille flux in the equivalent pore reads:

$$\langle J^i_{H-P} \rangle_{\hat{\Omega}} = -\tau \frac{r^2}{8\nu_l}\nabla \langle p_l \rangle_{\hat{\Omega}} \tag{63}$$

where $r$ is the radius of the pore.

Setting Equations (62) and (63) equal, the tortuosity-connectivity parameter can be written as a function of the permeability function:

$$\tau(r) = \frac{8\hat{k}_l}{r^2} \tag{64}$$

The tortuosity-connectivity for a fully saturated homogeneous material can be calculated from Equation (64) with $r = R$, where $R$ is the radius of the saturated equivalent pore associated to a fully saturated REV:

$$\tau(R) = \frac{8\hat{k}_l}{R^2} \tag{65}$$

The radii $r$ and $R$ are related according to the following expression:

$$r^2 = R^2 \frac{\phi_l}{\phi} \tag{66}$$

where $\phi$ is the porosity of the REV.

The tortuosity parameter can now be expressed as a function of the pore water content:

$$\tau(\phi_l) = \frac{8\hat{k}_l}{R^2}\frac{\phi}{\phi_l} \tag{67}$$

The permeability is often expressed as the product of an intrinsic permeability coefficient $\hat{K}$ (m$^2$) and a permeability function relative to the $l$-phase $\hat{k}_{rl}$, which takes values between 0 and 1. Equation (67) can then finally be rewritten as:

$$\tau(\phi_l) = \tau(\phi)\frac{\phi}{\phi_l}\hat{k}_{rl}(\phi_l) \tag{68}$$

*5.2. Calculation of the Tortuosity-Connectivity Parameter*

Experimental measurements of the tortuosity-connectivity would require determining the ttortuosity-connectivity for various values of the pore water content, which should be distributed uniformly within the material. Methods to obtain a uniformly distributed pore water content in concrete can be found in research work [28–30].

According to Equation (68), the tortuosity-connectivity parameter of the porous medium can be calculated if its permeability is known. The permeability is often modelled by means of the van Genuchten model:

$$\hat{k}_l(\phi_l) = \hat{K}\sqrt{\frac{\phi_l}{\phi}}\left(1 - \left[1 - \left(\frac{\phi_l}{\phi}\right)^{\frac{1}{e}}\right]^e\right) \tag{69}$$

where $e$ is a material constant.

The expression for the tortuosity-connectivity parameter then becomes:

$$\tau(\phi_l) = \tau(\phi)\sqrt{\frac{\phi}{\phi_l}}\left(1 - \left[1 - \left(\frac{\phi_l}{\phi}\right)^{\frac{1}{e}}\right]^e\right) \tag{70}$$

In order to calibrate this model with experimental results, Equation (70) was used in this study, where the material constant $e$ and the tortuosity-connectivity in saturated conditions $\tau(\phi)$ were measured for four different concretes, by fitting Darcy's law to experimentally obtained data.The procedure consists of solving the equation of mass flux of the pore solution, based on the Darcy flow, which calculates a profile of water in the pores (see [5] for more details). Then this profile has been integrated over the spatial domain to obtain the total mass of water penetrated in thin concrete samples of 2 mm thickness. As the Darcy flow depends on the tortuosity-connectivity model Equation (70), this mass comparison has allowed us to determine the values of $e$ and $\tau(\phi)$. In order to determine the hygroscopic properties of concrete, it is necessary to measure the intrinsic permeability coefficient of the material, as well as the relation between relative humidity and water content within the material. The permeability coefficients of the materials were indirectly obtained by absorption tests. The relation between relative humidity and water content was obtained for each material by means of absorption and desorption tests at different relative humidities. The absorption and desorption tests were performed according to the instructions of the standard test method EN ISO 12571 [31]. The test consists basically in placing a concrete sample in some constant environment (constant temperature and constant relative humidity) and weigh the sample when it is entirely in equilibrium with its surroundings. The greater the concrete samples, the longer it takes them to reach equilibrium. Therefore, very thin samples (aproximately 2 mm) were used.

The four concretes were designed in acordance with usual mixes used in agressive environments to improve the impermeability and resistance against penetration of ionic species. Concrete dosages are referred to as *mat1*, *mat2*, *mat3* and *mat*. The cement type used for materials *mat1* (without additions), *mat2* (with addition of 20% of fly ash) and *mat4* (with addition of of 10% of silica fume) is CEM I 42.5 R/SR according to the EN 197-1 Standard [32]. The cement employed for material *mat3* contains 66% blast furnace slag and its type is known as CEM III/B 42.5 L/SR, in agreement with the EN 197-1 Standard [32]. The properties of the materials are described in detail in [33]. The dosages of the materials are given in Table 1.

**Table 1.** Dosages of the studied materials.

| Material | Dosages (kg/m³) | | | |
|---|---|---|---|---|
| | *mat1* | *mat2* | *mat3* | *mat4* |
| Cement | 380 | 357 | 380 | 304 |
| Water | 171 | 194 | 171 | 154 |
| Fly ash | 0 | 76 | 0 | 0 |
| Silica fume | 0 | 0 | 0 | 38 |
| Aggregate | 787 | 770 | 787 | 800 |
| Sand | 1022 | 966 | 1022 | 1067 |
| Superplasticizer * | 0.97 | 0.70 | 1.30 | 1.80 |

* % of cement weight.

The tortuosity-connectivity function of the materials, obtained from the experimental results, is plotted in Figure 4. It is shown that the tortuosity-connectivity parameter decreases significantly for lower pore water contents, meaning that the mass fluxes are significantly reduced in the direction of the net mass flux. In saturated conditions, the tortuosity-connectivity functions reach their maxima. It can be observed that the tortuosity-connectivity undergoes hysteresis, which is due to the fact that the pore water can get trapped in ink-bottle shaped pores when the concrete is subjected to drying conditions [5]. It is worth noting that this phenomenon has been found significantly greater for *mat1* and *mat2*, probably due to a significantly higher amount of bottleneck pores, being the two mixtures with the higher total porosity as can be seen in [33]. Similarly, *mat1* and *mat2* presented the greater value of the tortuosity-connectivity parameter. In the case of *mat3* and *mat4*, the tortuosity-connectivity parameter obtained was considerably lower, with also a lower total porosity and average pore size [33] than the rest. This is coherent with the results obtained by [33] for differential thermal analysis, mercury intrusion porosimetry, water and gas permeability and mechanical properties of The materials, in which the mixtures could be divided into two groups according to its similar behaviour, presenting better performance for mixtures *mat3* and *mat4*.

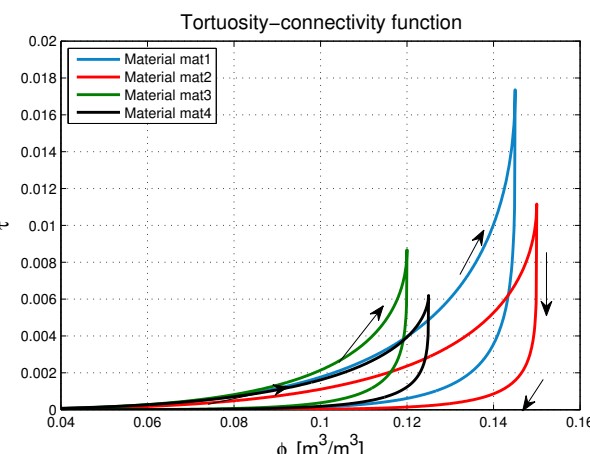

**Figure 4.** The tortuosity as a function of the pore water content.

Using these tortuosity-connectivity functions, the diffusion coefficient for ionic species $\tau(\phi_l)\phi_l D_i$ can be obtained. Such coefficients were measured for chloride ions and for several pore water contents. As can be deduced from Figure 4, the diffusion coefficient $\tau(\phi_l)\phi_l D_i$ increases with the pore water content.

### 5.3. Closure Equations

As shown in [5], the terms in Equation (61) related to migration and chemical activity can generally be ignored. For the sake of simplicity, the same assumptions are made in this paper, yielding:

$$\frac{\partial}{\partial t}\left(\phi_l \langle u_i \rangle_{\hat{\Omega}_l}\right) + \nabla.\left(\langle u_i \rangle_{\hat{\Omega}_l} \frac{k_l}{\nu_l} \tau \nabla \langle p_c \rangle_{\hat{\Omega}_l}\right)$$
$$= \nabla.\left(\phi_l D_i \tau \nabla \langle u_i \rangle_{\hat{\Omega}_l}\right) + \phi_l \langle r_i \rangle_{\hat{\Omega}_l} \tag{71}$$

The capillary pressure $\langle p_c \rangle_{\hat{\Omega}_l}$ was modelled by means of the capillary model proposed in [5]:

$$\langle p_c \rangle_{\hat{\Omega}_l} = A(atan(a[b - \phi_l]) + atan(a[\phi - b])) \tag{72}$$

where $A$ ($MPa$) is the capillary modulus, and $a$ and $b$ are material constants.

The reaction term $\langle r_i \rangle_{\hat{\Omega}_l}$ was modelled by means of the Langmuir adsorption equation, which in this case can be expressed as [5]:

$$\phi_l \langle r_i \rangle_{\hat{\Omega}_l} = \phi_l \langle \hat{u}_i \rangle_{\hat{\Omega}_l} \hat{\Omega}_l^{-1} k_{eq} \frac{\partial}{\partial t}\left(\frac{\phi_l \langle u_i \rangle_{\hat{\Omega}_l}}{\hat{\Omega}_l + k_{eq}\phi_l \langle u_i \rangle_{\hat{\Omega}_l}}\right) \tag{73}$$

where $k_{eq}$ (m$^3$/kg) is a constant which describes the equilibrium between the free ions and the bound ions, while $\langle \hat{u}_i \rangle_{\hat{\Omega}_l} \hat{\Omega}_l^{-1}$ (kg/m$^3$) is the maximum density of reactants in volume $\hat{\Omega}_l$.

Substituting Equations (69), (70), (72) and (73) into Equation (71), the closure equations are obtained. The closure equation for chloride ions was solved by means of the finite element method. Some results are shown in the next section.

## 6. Numerical Simulations of Experimental Chloride Profiles

### 6.1. Overview of the Experimental Programme

Once the tortuosity-connectivity parameter had been calibrated with concretes widely studied by the authors and its good fit had been verified to adequately simulate the experimental results obtained, the model was tested by comparing the numerical simulations with experimental chloride profiles obtained from two different concretes. The latter were obtained from two different mixtures (referred to as *mat5* and *mat6*) selected to represent the two types of behaviour observed in the previous experimental campaign. This way, in the new experimental campaign the reference concrete *mat5* is a mixture without additions representing the group *mat1* and *mat2*, and *mat6* is a mixture containing 10% of silica fume, representing the group with the best general durable perfomance *mat3* and *mat4*. Both dosages were based on Portland cement. The water-cementitious materials ratios were 0.40 and 0.45 for the concrete with silica fume. More details are given in Table 2. These materials were tested under several boundary conditions in order to experimentally simulate half a year of service life of a concrete subjected to high mountain environment with melting salts, which is a highly aggresive environment with chlorides. The initially fully saturated concrete samples were subjected to five different aggressive environments for 180 days. The first 75 days, the samples were exposed to a chloride solution of 100 g/L at a temperature of 1 °C. This phase simulated the winter season in which chloride solutions were sprayed over the roads and concrete structures in order to avoid the forming of ice patches on the road surface. As a result, the chloride ions started to penetrate into the concrete. On day 76, a thin layer (2 cm) of fresh water was poured onto the exposed surface at a temperature of 10 °C, and was removed 30 days after. This phase corresponded to a rainy spring. The rain eliminated chlorides from the concrete structures. In order to simulate a drier summer, for the next 60 days, the concrete was dried at a relative humidity of 75% and temperatures of $T = 16\,°C$ and $T = 5\,°C$ (30 days each). During that season, the chloride ions were transported towards the surface, where they started to precipitate. Finally,

the samples were exposed once more to a chloride solution of 100 g/L at temperature 3 °C for 15 days. A schematic representation is given in Table 3.

**Table 2.** Dosages of the studied materials.

| Dosages (kg/m$^3$) | | |
| --- | --- | --- |
| **Material** | *mat5* | *mat6* |
| Cement | 400 | 320 |
| Water | 160 | 180 |
| Silica fume | 0 | 40 |
| Ratio water/cementitious materials | 0.40 | 0.45 |
| Aggregate | 970 | 1015.73 |
| Sand | 846 | 861.08 |
| Superplasticizer * | 1.5 | 1.5 |

* % of cement weight.

**Table 3.** The aggressive environments.

| Aggressive Environments | | | |
| --- | --- | --- | --- |
| **Days** | **Chloride Concentration (g/L)** | **Relative Humidity (%)** | **Temperature (°C)** |
| 1–75 | 100 | 100 | 1 |
| 76–105 | 0 (at day 76) | 100 | 10 |
| 106–135 | - | 75 | 16 |
| 136–165 | - | 75 | 5 |
| 166–180 | 100 | 100 | 3 |

*6.2. Numerical Modelling*

In order to account for the relative humidity and temperature, the transport model was coupled to differential equations based on Darcy's law (to solve for the pore water content) and on Fourrier's law (to solve for temperature). The coupled differential equations were solved by means of the finite element method. A detailed description of the formulation can be found in [5]. The model was then calibrated for the materials as described in [5].

The numerical results versus the experimental chloride profiles are plotted in Figures 5–7 for material *mat5* and Figures 8–10 for material *mat6*. The vertical axis represents the chloride profiles expressed in kg/m$^3$ of concrete, while the horizontal axis represents the penetration depth expressed in mm. It can be observed that a good fit was obtained as numerical simulations reproduced the experimental results with a significantly good fit. The capillary model enabled modelling of the pore water flow, accounting for the properties of the microstructure of the material. The microstructure was characterized by the tortuosity-connectivity parameter which was quantified in function of the pore radii for different materials.

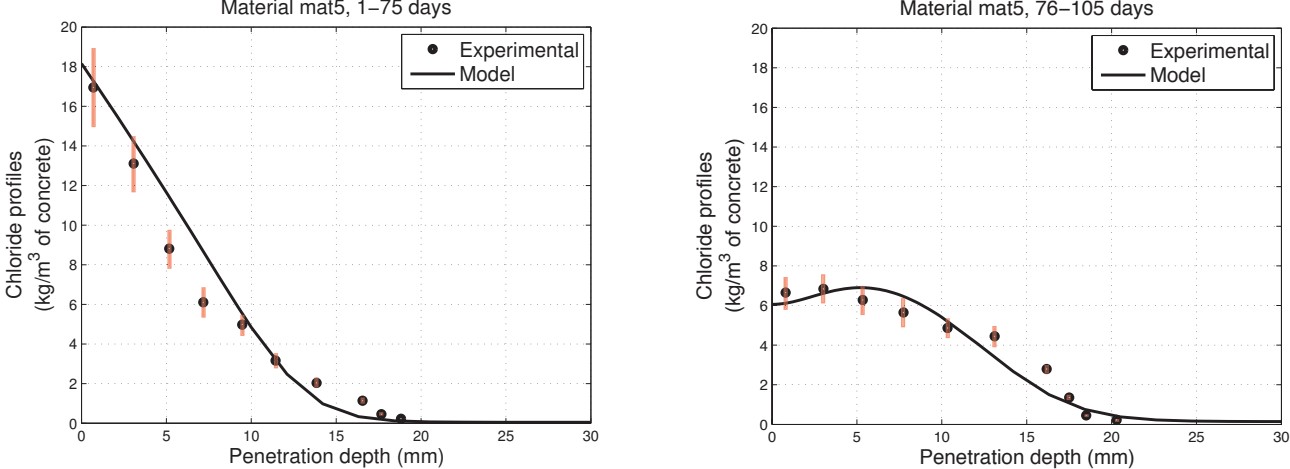

**Figure 5.** Experimental versus numerical solution for material *mat5* and environments 1 (1–75 days) and 2 (76–105 days).

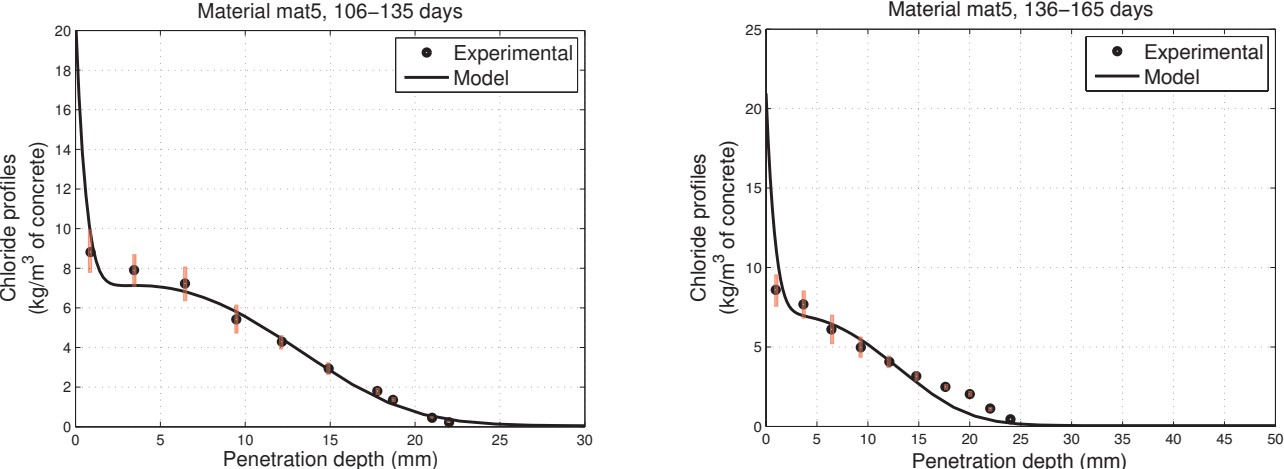

**Figure 6.** Experimental versus numerical solution for material *mat5* and environments 3 (106–135 days) and 4 (136–165 days).

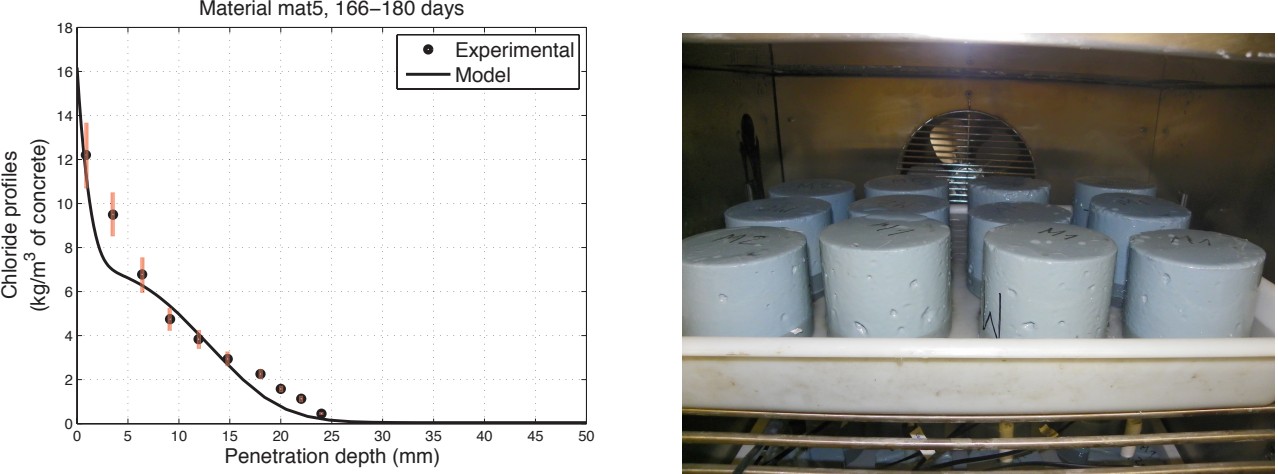

**Figure 7. Left**: experimental versus numerical solution for material *mat5* and environment 5 (166–180 days). **Right**: concrete samples exposed to water.

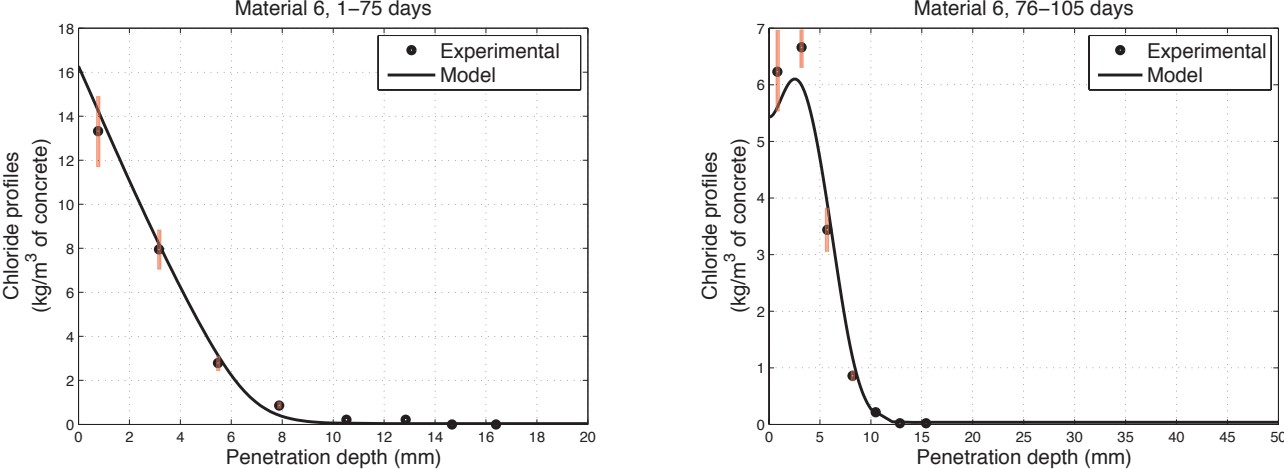

**Figure 8.** Experimental versus numerical solution for material *mat6* and environments 1 (1–75 days) 2 (76–105 days).

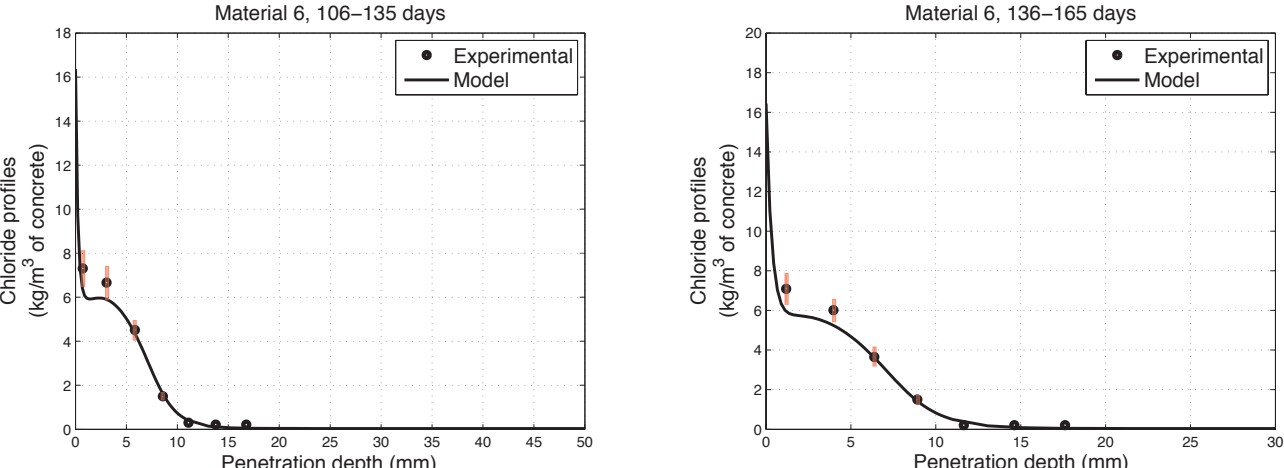

**Figure 9.** Experimental versus numerical solution for material *mat6* and environments 3 (106–135 days) and 4 (136–165 days).

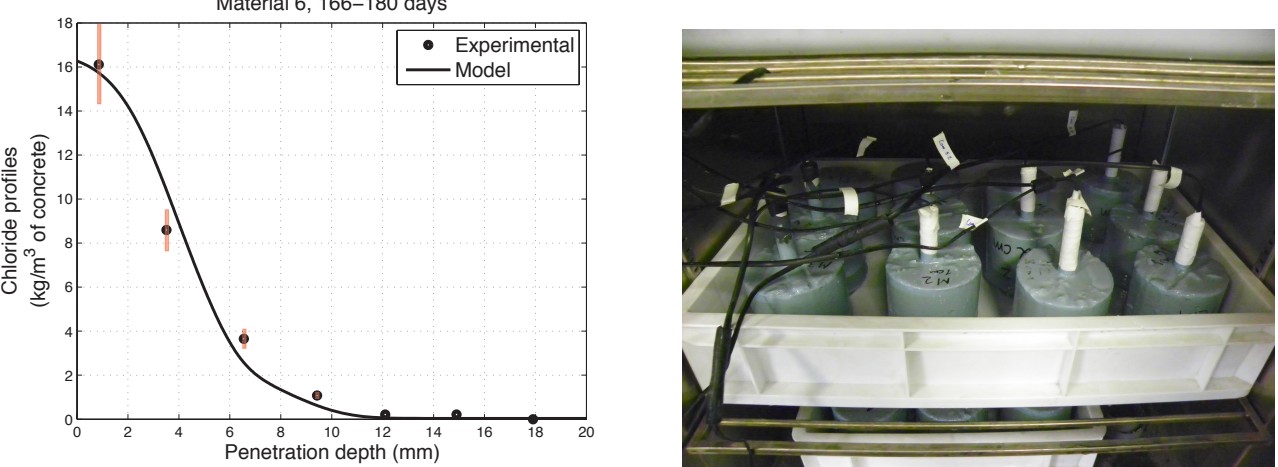

**Figure 10. Left**: experimental versus numerical solution for material *mat6* and environment 5 (166–180 days). **Right**: measurements of the relative humidity within the concrete samples.

### 7. Conclusions

The study carried out in this work led to the following conclusions:

1. In order to obtain the transport equations at the macroscopic scale, the microscopic equations were integrated over the REV by means of the well known averaging technique.
2. It was shown that the dispersion terms which arose from the averaging procedure can be ignored.
3. The surface integrals which arose from the averaging technique were interpreted. It was shown not only how they are related to the tortuosity-connectivity of the porous network, but also how they influence the mass fluxes of the ionic species.
4. The transport equations in the REV were compared with the equations in an equivalent pore, oriented in the direction of the net mass flux.
5. An explicit expression of the tortuosity-connectivity parameter was obtained by comparing Darcy's law and the Hagen–Poiseuille equations for the equivalent pore.
6. The proposed model accurately fits the results obtained from experimental tests performed on concretes in non-saturated conditions.

**Author Contributions:** Conceptualization, M.F., E.R., A.M. and J.C.G.; methodology, M.F., E.R., A.M. and J.C.G.; software, M.F.; validation, M.F., E.R., A.M., J.C.G.and J.B.; formal analysis, M.F., E.R., A.M. and J.C.G.; investigation, M.F., E.R., A.M., J.C.G. and J.B.; resources, E.R., A.M. and J.C.G.; data curation, M.F. and J.B; writing—original draft preparation, M.F., E.R. and J.C.G.; writing—review and editing, M.F., E.R., A.M. and J.C.G.; visualization, M.F., E.R., A.M. and J.C.G.; supervision, E.R., A.M. and J.C.G.; project administration, E.R., A.M. and J.C.G.; funding acquisition, M.F., E.R., A.M. and J.C.G. All authors have read and agreed to the published version of the manuscript.

**Funding:** This research was funded by Spanish Ministry of Economy and and Competitiveness of Spain by means of the Research Projects PID2019-108978RB-C31 and RTI2018-100962-B-100.

**Institutional Review Board Statement:** Not appplicable.

**Informed Consent Statement:** Not appplicable.

**Data Availability Statement:** The data presented in this study are available on request from the corresponding author.

**Conflicts of Interest:** The authors declare no conflict of interest.

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
