# Peer review of "On the Tortuosity-Connectivity of Cement-Based Porous Materials"

_applsci, doi:10.3390/app11135812_

Round 1

Reviewer 1 Report

In my opinion the manuscript "applsci-1238951" is suitable for publication in the journal of "Applied Sciences " after minor revision. Authors report a new macroscale modelling approach to predict chloride ion transport in the concrete which is useful to increase the durability of  reinforced concrete with steel.

There are few minor spelling and language errors which should be corrected. Selected list of spelling errors:

Page 18, point 5 of the conclusion:  ttortuosity 

Page 12, few lines before table 1:  reference to the standard is missing :RC-16 Standard (? )

Page 2, line 3:  citation number is replaced with "?" "Bourdette et al. (? ) "

----

Suggestion: the introduction can be amended by comparing/proving the macro-modeling with mico/nano-models . 

Author Response

In my opinion the manuscript "applsci-1238951" is suitable for publication in the journal of "Applied Sciences " after minor revision. Authors report a new macroscale modelling approach to predict chloride ion transport in the concrete which is useful to increase the durability of  reinforced concrete with steel.

There are few minor spelling and language errors which should be corrected. Selected list of spelling errors:

Page 18, point 5 of the conclusion:  ttortuosity    Corrected

Page 12, few lines before table 1:  reference to the standard is missing :RC-16 Standard (? )  corrected

Page 2, line 3:  citation number is replaced with "?" "Bourdette et al. (? ) " Corrected

----

Suggestion: the introduction can be amendd by comparing/proving the macro-modeling with mico/nano-models . 

Thank you very much for your suggestion. A comment and a reference have been added in the Introduction following your recommendation.

The authors really appreciate the comments of the reviewer for improving the quality of the manuscript.

Reviewer 2 Report

  1. Pleas fill the reference in Page 2, line Bourdette et al. (? ) found a notably…..
  2. Explain why the value of the diffusion coefficient should be selected by the equation (57), and the physical meaning of the dimensionless group tsimDi/d2.
  3. Fig. 2 and fig.3 show that tortuosity-connectivity increases the average speed of the ions. Would you explain why the tortuosity-connectivity will increase the speed of ions in liquid phase? Does the tortuosity-connectivity enhance diffusive mass flux, the advective mass flux , the mass flux related to migration or the mass fluxes due to temperature gradients and chemical activity?
  4. Explain in the experiment how to obtain the tortuous-connectivity function of the material in Figure 4.

Author Response

Comments and Suggestions for Authors

  1. Pleas fill the reference in Page 2, line Bourdette et al. (? ) found a notably….. Corrected
  2. Explain why the value of the diffusion coefficient should be selected by the equation (57), and the physical meaning of the dimensionless group tsimDi/d2.

56

Using tsimDi/din the equation 56 results in a dimensionless version of the equation. Since the values of d and tsim are the same for both pores, the use of equation 57 is not necessary but makes the procedure easier. The diffusion coefficient is chosen in such a way that the square length traveled by the ion during the total simulation time is equal to the square pore length.

A couple of wording changes have been made in the text in order to help improve the understanding of the procedure.

  1. Fig. 2 and fig.3 show that tortuosity-connectivity increases the average speed of the ions. Would you explain why the tortuosity-connectivity will increase the speed of ions in liquid phase? Does the tortuosity-connectivity enhance diffusive mass flux, the advective mass flux , the mass flux related to migration or the mass fluxes due to temperature gradients and chemical activity?

Fig.2 and Fig. 3 illustrate what happens with the diffusive flux in a straight pore versus a pore with difficult geometry. This analysis has been done only for the diffusion flux. For the sake of simplicity, as stated later in section 5.3, in this study fluxes related to migration and chemical activity are ignored. This assumption is made in this paper based on previous works (M Fenaux, E Reyes, JC Gálvez, A Moragues “Modelling the transport of chloride and other ions in cement-based materials” Cement and Concrete Composites 97, 33-42, 2019) in which the contribution to the flux of these mechanisms is studied in detail, concluding that the contribution is significantly lower than diffusion in both cases, and therefore can be neglected in this analysis. In order to clarify this section, the last sentence before section 4.6.1, has been changed to "The second term corrects the mass flux by accounting for the complex geometry of the porous medium as shown below for the diffusion flux. For the sake of simplicity, the fluxes related to migration and chemical activity are neglected (ref. 27, M Fenaux et al. 2019).”

In the case of diffusion, the velocity increases probably due to the bottleneck that is formed in the pore geometry. In this area the ions move closer together, resulting in steeper concentration gradients.

  1. Explain in the experiment how to obtain the tortuous-connectivity function of the material in Figure 4.

The procedure consists of solving the equation of mass flux of the pore solution, based on the Darcy flow, which calculates a profile of water in the pores (see [5] for more details). Then this profile has been integrated over the spatial domain to obtain the total mass of water penetrated in thin concrete samples (2 mm thickness. As the Darcy flow depends on the tortuosity-connectivity model (equation 71 of the paper), this mass comparison has allowed to determine the values of the material constants e and τ(φ).

In order to determine the hygroscopic properties of concrete, it is necessary to measure the intrinsic permeability coefficient of the material, as well as the relation between relative humidity and water content within the material. The permeability coefficients of the materials were indirectly obtained by absorption tests. The relation between relative humidity and water content was obtained for each material by means of absorption and desorption tests at different relative humidities.

The absorption and desorption tests were performed according to the instructions of the standard test method EN ISO 12571 (31). The test consists basically in placing a concrete sample in some constant environment (constant temperature and constant relative humidity) and weigh the sample when it’s entirely in equilibrium with its surroundings. The greater the concrete samples, the longer it takes them to reach equilibrium. Therefore, very thin samples (∼ 2mm) were used.

The authors really appreciate the comments of the reviewer for improving the quality of the manuscript.

Reviewer 3 Report

In this manuscript authors show characterization of porous concrete. The information shown in the manuscript are interesting. The characterization methods are well chosen. However, there are few points which should be improved:

  1. The title should be more precise and contains information about the investigated materials.
  2. Statistical data: please add error bars in fig 6, 7, 8, 9 and 10.
  3. The references: authors could expand the list.

My recommendation is: major revision

Regards

Author Response

In this manuscript authors show characterization of porous concrete. The information shown in the manuscript are interesting. The characterization methods are well chosen. However, there are few points which should be improved:

  1. The title should be more precise and contains information about the investigated materials

Thank you very much for your suggestion. The title have been changed following your recommendation

  1. Statistical data: please add error bars in fig 6, 7, 8, 9 and 10.

Thank you for your comment. Error bars have been incorporated.

  1. The references: authors could expand the list.

The authors thank again for the reviewer's comment. Three more works have been referenced in the introduction section. In addition, two other references have been introduced throughout the text

The authors really appreciate the comments of the reviewer for improving the quality of the manuscript.

Round 2

Reviewer 2 Report

 This is an interesting paper.

Reviewer 3 Report

After revision the manuscript could be accept

Regards